# Clinical and Sonographic Evaluation of Postmenopausal Bleeding (PMB) Followed by Diagnostic and/or Therapeutic Hysteroscopy and Guided Biopsy in Jordanian Hospitals

**DOI:** 10.3390/medicina56040147

**Published:** 2020-03-25

**Authors:** Abu-Azzam Omar, Abufraijeh Seham, Ahlam Mahmoud Al-Kharabsheh, Emad Alshara, Amer Mahmoud Sindiani, Omar Hamdan, Imene Ghoul, Hamzeh Mohammad Alrawashdeh

**Affiliations:** 1Department of Obstetrics and Gynecology, College of Medicine, Mutah University, al-Karak 61710, Jordan; sehamabufraijeh@yahoo.com (A.S.); ahlamalkharabsheh@gmail.com (A.M.A.-K.); 2Department of Obstetrics and Gynecology, Royal Medical Services, Amman 00962, Jordan; emadalsharu@hotmail.com; 3Department of Obstetrics and Gynecology, College of Medicine, Jordan University of Science and Technology, Irbid 22110, Jordan; obs2gyn@yahoo.com; 4College of Medicine, Mutah University, al-Karak61710, Jordan; Omarhamdan321988@gmail.com; 5Department of Pediatric, Ibn-Alhaytham Hospital, Amman 11194, Jordan; Emanghoul@hotmail.com; 6Department ofOphthalmology, Ibn-Alhaytham Hospital, Amman 11194, Jordan; dr_hmsr@yahoo.com

**Keywords:** bleeding, postmenopausal, sonographic evaluation, women

## Abstract

*Background and Objectives:* The goal of this study was to evaluate the clinical sonographic evaluation of postmenopausal bleeding (PMB) followed by diagnostic and/or therapeutic hysteroscopy and guided biopsy in Jordanian hospitals. *Materials and Methods:* A retrospective multi-centric study was performed in hospitals in Al-Karak and Amman from 2014–2016. The study recruited 189 cases to evaluate the aetiology of postmenopausal bleeding. Atrophic endometrium was observed as a major cause of postmenopausal bleeding according to histopathology. The cases were also distributed according to parity in which nulliparous patients were observed. *Results:* Hysteroscopy was observed to be effective for the diagnosis of postmenopausal bleeding. *Conclusion:* There is a need to assess more approaches for the diagnosis of postmenopausal bleeding among women.

## 1. Introduction

Postmenopausal bleeding (PMB) is the most commonly observed conditionin gynaecological practice. The prevalence rate of endometrial cancer among women has been shown to be10% [1]. The identification of women with a low or high risk of endometrial cancer can be done by transvaginal sonographic measurement of endometrial thickness [1]. Women are determined to beat a higher risk of endometrial cancer if the endometrium is thickened, and in such case, endometrial sampling is implemented. Among women with thickened endometrium and benign endometrial sampling with PMB, operative hysteroscopy is unable to lessen recurrent bleeding [2].

In cases where insistent bleeding is not a symptom, advanced diagnostic examination along with expectant management techniques are used for diagnosing focal lesion. Specifically, hysteroscopy or saline infusion sonography (SIS) is performed in order toobtain the most precise and accurate outcomes. The clinicians who recommendmore testing do sobecause undiagnosed endometrial polyps are considered to beresponsible for recurrent vaginal PMB. A cohort study that evaluated PMB among women who had an endometrial biopsy alone, polypectomy after detection at hysteroscopy, or hysteroscopy with biopsy reported no difference in recurrent bleeding [3]. In light ofthe lack of established literature, this study performed a multicenter randomized controlled trial to investigate the PMB followed by diagnostic and/or therapeutic hysteroscopy and guided biopsy in Jordanian hospitals.

An increased risk of endometrial cancer has been observed among women experiencingpostmenopausal bleeding and an endometrial thickness greater than 4 mm at initial workup [4]. As such, there is a need to identify the diagnostic approaches for the evaluation of postmenopausal bleeding implemented in Jordanian hospitals in order to reduce the risk and associated complications of postmenopausal bleeding among women. The study has effectively assessed the atrophic endometrium, endometrial hyperplasia, endometrial polyp, sub-mucous fibroid, and endometrial cancer in terms of transvaginal sonography (TVS) sensitivity, TVS specificity, HYST sensitivity, and HYST specificity.

The study has contributed to identifying the risks associated with post-menopausal endometrial pathology and the clinical and sonographic evaluation of post-menopausal bleeding. The effectiveness of hysteroscopy and guided biopsy has been investigated for the diagnosis of postmenopausal bleeding. The following sections examine the effectiveness and efficiencies of the diagnostic approach of postmenopausal bleeding under different conditions and factors. The literature review has been presented to show the previous work conducted to assess postmenopausal bleeding and endometrial thickness under different circumstances. Furthermore, the methodology employed to achieve the aim has been presented, and results analyzed from the survey have also been shownin order to outline the clinical and sonographic evaluation of postmenopausal bleeding (PMB) followed by diagnostic and therapeutic hysteroscopy and guided biopsy in Jordanian hospitals.

## 2. Literature Review

A study conducted by Smith et al. [5] reported that recurrent PMB results in less probability of malignant and premalignant endometrial disease, although PMB is caused due to endometrial polyps in one infour women. The first-line investigation should include a high-accuracy test (such as saline infusion sonography and outpatient hysteroscopy) for the women with recurrent PMB in order to enable diagnosis of focal diseases. Women with PMB must be timely examined in order to eliminate endometrial carcinoma. Transvaginal ultrasonography can be considered a reasonable first-line approach along with invasive sampling, whichis necessary when the ultrasonographic endometrial thickness is above 4 mm [6]. However, non-malignant conditions have been observed among 90% women with PMB. Therefore, women who havean increased risk of cancer must be further identified. Individual patient characteristics are required in order toattain this aim and allow for a more precise evaluation strategy.

Postmenopausal women with bleeding must be managed and investigated accordingly. One of the useful non-invasive techniques for preliminary evaluation of these women is ultrasound. Endometrial biopsy is recommended for symptomatic women with the symptom of thickened persistentendometrium and endometrium bleeding. An incidental result of the endometrial pathology in asymptomatic postmenopausal women poses a clinical management dilemma [7].

Histological findings along with hysteroscopic images were used to diagnose focal thickening. Four cases were related to atypia-free endometrium, three cases were related to endometrial hyperplasia, two cases were related with polyps, and one case was related with a proliferative endometrium.

According to the study conducted by Baracat et al. [8], the cut-off pointfor menopausal endometrial thickness in ultrasound is 4 to 5 mm. It has beenobservedthat if endometrial echo is 5mm or above during ultrasound then it is a sign of having no endometrial disease. It usually indicates that the ultrasound could not exclude the disease, and an investigation must be carried out by hysteroscopy. The study further investigated the histological findings in patients with hysteroscopic images. It suggested the presence of focal thickening, which is commonly followed by proliferative endometrium, polyps, endometrial hyperplasia, and atypia-free endometrium.

One of the most suitable methods to examine the uterine cavity of women withendometrial thickening (without or with symptoms) is hysteroscopy. It is an examination that may be accomplished in an outpatient setting. It may also permit a direct view of the endometrium and biopsies, which can be considered an advantage that enhances diagnostic accuracy, specifically in regard tofocal lesions [8]. There is limited value of routine screening with endometrial biopsy in asymptomatic women on tamoxifen [7]. All the spotted and abnormal bleeding must be investigated, but pipelle endometrial biopsyrarely provides useful diagnostic knowledge among women treated with tamoxifen. Therefore, it has been suggested that symptomatic women who have thickened endometrium must be investigated with a targeted biopsy and hysteroscopy. The thickened endometriummainly occurs due to the tamoxifen-induced subepithelial stromal hypertrophy [7].

A study by Tinelli et al. [9] was conducted to determinethe diagnostic accuracy of transvaginal ultrasonography and hysteroscopy for endometrial pathology diagnosis among postmenopausal women with abnormal uterine bleeding. It was reported that hysteroscopy is a more accurate diagnostic method for the endometrial pathology detection than transvaginal ultrasonography. Transvaginal ultrasonographyhas enhanced specificity and must be considered with an endometrial thickness of more than 4 mm for patients with abnormal uterine bleeding. Hysteroscopy presents greater effectiveness in the diagnosis of the endometrium, particularly in regard tofocal abnormalities, which are improbably identified by ultrasonography and must be reflected in the conditions of abnormal uterine bleeding with an endometrium thickness of 4 mm on ultrasonography. Among women presenting suspicious and abnormal lesions, it is essential to implement hysteroscopy with eye-directed biopsy due to some conditions of endometrial carcinoma.

A study by Dueholm et al. [10] compared and evaluated the inter-observer variation within endometrial pattern identification with hysteroscopy, transvaginal sonography, and gel infusion sonography regarding endometrial pathology diagnosis. It has been observed that the agreement of observers regarding HYST and TVS was consistent and reliable for the diagnosis of normal endometrium, whereas, HYST, TVS failed for the diagnosis of cancer. Low agreement between observers was observed among the patients with cancer or hyperplasia when HYST was used in the presence of additional polyps. This alarming situation brings to attention the need for the improvement of systematic methods and reliability in endometrial pattern identification.

## 3. Materials and Methods

This retrospective multi-centric study was performed in Al-Karak teaching hospital and private hospitals in Al-Karak and Amman between the years 2014 and 2016. The study recruited 189 cases to examine the aetiology of postmenopausal bleeding (PMB). This study was approved by the ethical committee (IRB) (Ethical code number: 2020150, February 26th, 2016). The significance of hysteroscopy in the evaluation of pathogenic factors relating todiagnosis after transvaginal sonography (TVS), hysteroscopy, and histopathologic diagnosis, as well as thefeasibility of conservative management with hysteroscopy in PMB, were assessed. The study recruited patients of different ages. The causes of postmenopausal bleeding according to histopathology were noted. The causes included atrophic endometrium, exogenous estrogen, endometrial cancer, endometrial polyps, and endometrial hyperplasia. Body mass index (BMI) was obtained foreach patient. The cases were distributed according to parity. The comorbid condition included the number of women with carcinoma endometrium and the number of women with PMB. Correlationswere observed between endometrial thickness (ET) andTVS and PMB. The sensitivity and specificity of TV/US and hysteroscopy for diagnosing endometrial pathologies causing PMB were evaluated.

## 4. Results

The study was conducted on 189 cases, and the results obtained included 5% patients who were less than 45 years, 17% who were 45–49 years old, 52.9% who were 50–55 years old, and 25.1% who were more than 55 years old. The main causes of postmenopausal bleeding according to histopathology were also assessed. Atrophic endometrium was found to bea major cause of postmenopausal bleeding according to histopathology (67%). Exogenous estrogen was not observed to be affecting the postmenopausal bleeding (0%), as it has an inverse relation with postmenopausal bleeding. Endometrial cancer (11.6%), endometrial polyps (15%), and endometrial hyperplasia (6.4%) were observed as some causes of postmenopausal bleeding. The body mass index (BMI) values of the patients were estimated, which showed that 6.6% patients had a BMI less than 18.5, 30% patients had a BMI between 18.5 and 24.9, 50% patients had a BMI between 25 and 29.9, and 13.3% patients had a BMI more than 30. The cases were also distributed according to parity, in which 30% of patients nulliparous, 51.5% were primipara, and 18.5% were multipara. The comorbid conditions among patients recruited in the study can be observed in Table 1 and Table 2.

## 5. Discussion

Transvaginal ultrasound is being increasingly used to measure the endometrial thickness as a first-line method to assess patients withvaginal bleeding issues. A study by Patel et al. [11] also aimed to investigate the association of the outcomes of transvaginal ultrasound and the histopathologic investigation by determininga threshold that can reliably exclude carcinoma. Women ages 55 years and above who presented postmenopausal bleeding and experienced transvaginal ultrasound within thirty days of the endometrial biopsywere included in the study. Patients were grouped on the basis of diseases they were suffering from. The results showed that low endometrial thickness was present and that a thickness of 3 mm to 4 mm represented an ideal threshold for maximizing the sensitivity of the analysis. Sensitivity was observed to be maximized to 96.9% by utilizing a threshold of 3 mm. Endometrial pathology can be effectively indicated by the postmenopausal bleeding among women.

A study by Dueholm et al. [10] evaluated and compared the inter-observer variationin endometrial pattern recognition with transvaginal sonography, gel infusion sonography, and hysteroscopy. For the diagnosis, the observer agreement concerning both hysteroscopy and transvaginal sonography was reliable but poor for the diagnosis of normal endometrium. Agreement between researchers was perceived to be low among patients with cancer or hyperplasia when hysteroscopy was used in the presence of additional polyps. This study called attention to the requirement for systematic techniques to enhance reliability in endometrial pattern recognition.

The evaluation of postmenopausal bleeding is considered the most common procedure in gynecology settings. It has been shownthat the clinical and sonographic evaluation of PMB is directly associated with better diagnosis of impaired outcomes. A study carried out by Goldstein et al. [12] evaluated the role of sonography in women with postmenopausal bleeding. The study mentioned that postmenopausal bleeding (PMB) is a condition in which any vaginal bleeding occurs other than the expected cyclic bleeding among women. For this purpose, the study mentioned that therapeutic hysteroscopy plays a major role in the effective diagnosis of various conditions. GambadauroandGudmundsson [13] also mentioned that hysteroscopy is a significant tool for the diagnosis and evaluation of endometrial cancer. It is a fact that the early diagnoses of such outcomes through these techniques are directly associated with a reduced risk of complications. Sonographic evaluation along with the clinical evaluation, which is followed by therapeutic hysteroscopy and biopsy, is widely practiced to assure the delivery of quality healthcare to patients. It has been shownthat benign hysteroscopic findings are extremely common in IVF (in vitro fertilization) patients; however, the majority of the findings are identified through endometrial polyps, submucous fibroids, adhesions, or uterine anomalies.

The study also mentioned that the use of hysteroscopy can be harmful to women in certain cases; however, this aspect has not been evaluated in an efficient way. Thus, it is currently commonly held that clinical and sonographic evaluation of endometrial cancer can be achieved through therapeutic hysteroscopy along with the biopsy. In Middle Eastern countries, the delivery of healthcare requires proper technological advancements; therefore, the usage of therapeutic hysteroscopy and biopsy can be beneficial for patients in order to enhance their quality of life accordingly.

There was a limitation of this study in that hormonal profiling, as well as CT, and MRI imaging, could notbe evaluated either due to unavailability or financial costs.

## 6. Conclusions

The study identified risks related to the post-menopausal endometrial pathology and investigated the sonographic evaluation of post-menopausal bleeding. Hysteroscopy was found to be effective for the diagnosis of postmenopausal bleeding. The efficiencies and effectiveness of the diagnostic approach were evaluated under different circumstances.The clinical and sonographic evaluation of postmenopausal bleeding (PMB), followed by diagnostic and therapeutic hysteroscopy and guided biopsy in Jordanian hospitals, were successfully evaluated. Future studies may concentrate more on the diagnostic procedures of postmenopausal bleeding among women, as it is essential to evaluate the symptoms.

Key points from this study are as follows:The major cause of postmenopausal bleeding according to histopathology is atrophic endometrium.Hysteroscopy is an effective tool for the diagnosis of post-menopausal bleeding.The clinical and sonographic evaluations of postmenopausal bleeding, therapeutic hysteroscopy, and guided biopsy were investigated.

## Figures and Tables

**Table 1 medicina-56-00147-t001:** Comorbid conditions among patients recruited in the study. PMB: postmenopausal bleeding; BMI: body mass index.

Comorbid Condition	Women with PMB	Women with Carcinoma Endometrium
DM (Diabetes mellitus)	20%	5%
HTN (hypertension)	17%	25%
Hypothyroidism	7%	2%
BMI more than 30	14%	70%

**Table 2 medicina-56-00147-t002:** Sensitivity and specificity of TVS and hysteroscopy for diagnosing endometrial pathologies causing PMB. TVS: transvaginal sonography.

Diagnosis	TVS Sensitivity	TVS Specificity	HYST Sensitivity	HYST Specificity
Atrophic Endometrium	**88.7%**	**77.4%**	**96.3%**	**94.35%**
Endometrial Hyperplasia	**73%**	**94.25%**	**95.3%**	**98%**
Endometrial Polyp	**69%**	**94%**	**100%**	**100%**
Submucous Fibroid	**100%**	**100%**	**100%**	**100%**
Endometrial Cancer	**45%**	**88%**	**85.4%**	**96.1%**

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
