# Peer review of "Clinical and Sonographic Evaluation of Postmenopausal Bleeding (PMB) Followed by Diagnostic and/or Therapeutic Hysteroscopy and Guided Biopsy in Jordanian Hospitals"

_medicina, 2020, doi:10.3390/medicina56040147_

Round 1

Reviewer 1 Report

I read with great interest the Manuscript titled “Clinical and Sonographic Evaluation of Postmenopausal Bleeding (PMB) followed by Diagnostic and/or Therapeutic Hysteroscopy and Guided Biopsy in Jordanian Hospitals” (medicina-742392), which falls within the aim of Medicina.

In my honest opinion, the topic is interesting enough to attract the readers’ attention. Nevertheless, the authors should clarify some points and improve the discussion referring to relevant and novel key articles about the topic.

Authors should consider the following recommendations:

  • Manuscript should be further revised by a native English speaker.
  • Inclusion/exclusion criteria should be better clarified.
  • The Authors did not mention the sample size calculation for their study. It is essential to specify this data in order to guarantee an adequate significance of the results obtained by the Authors.
  • The authors have not adequately highlighted the strengths and limitations of their study. I suggest better specifying these points.
  • What are the actual clinical implications of this study and what does it add to the literature already available on the subject? Indeed, it is important to report the results obtained by the authors in the context of clinical practice and to adequately highlight what contribution this study adds to the literature already existing on the topic and to future study perspectives.
  • I could not find any information regarding the approval of the Institutional Review Board. Did author this approval before the study start?
  • I could not find any information regarding the informed consent of enrolled patients. Did author obtain informed consent for each patient? Conversely, this point may raise serious concern from the ethical point of view.
  • Previous published data suggested a potential difference of endometrial pathologies, especially endometrial cancer, in women with breast cancer treated with tamoxifen (TAM), aromatase inhibitors (AIs), or receiving no treatment (NT). Nevertheless, recent pieces of evidence suggest that TAM treatment does not seem to be associated with a higher rate of endometrial cancer in women with breast cancer compared with women treated with AIs or NT. I invite authors to discuss (at least briefly) this point, referring to: PMID: 31425735; PMID: 23599784.
  • I suggest discussing the recent innovations in the field of hysteroscopy and their clinical applications to endometrial biopsy. You may refer to: PMID: 31884079; PMID: 28108938.

Author Response

evaluation of the study. 2.  

Would you please specify which variables need means?

As the outcome variables of our study are dichotomous variable (TV/US, hysteroscopy) Namely Yes Vs. No.

3. Means 4. methods

I think he is asking about the means (ways or methods, like SPSS software program, statistical analysis, ROC CURVE for sensitivity and specificity)

5. Could the authors specify the correlation of these diagnostic findings with therapeutically approach and effectiveness ?

Could be!

But the main aim of the study was to evaluate the clinical Sonographicevaluation of Postmenopausal Bleeding followed by diagnostic and/or therapeutic hysteroscopy

6. Do the authors have any results of the screening of the hormonal profile or the CT scan of the patients to increrase the knowledge about them ?  

The requested results were unavailable right now as the study design was in retrospective manner from multicenter locations

Golden standard

2

1. Manuscript should be further revised by a native English speaker.

Done

2. Inclusion/exclusion criteria should be better clarified.

Exclusion criteria

1. Women taking hormonal replacement therapy 2. Obvious cause of bleeding from cervix and vagina 3. K/c/o bleeding dyscrasias 4. Anticoagulant therapy 5. Surgical menopause 6. TVS showing adnexal pathology

For each patient, detailed history was taken, which includes general medical history, menstrual and obstetric history, duration since menopause, severity and duration of PMB, history of gynecologic operations, drug intake and associated symptoms. A thorough general and systemic

3. The Authors did not mention the sample size calculation for their study. It is essential to specify this data in order to guarantee an adequate significance of the results obtained by the Authors.

The minimum sample size required for sensitivity and specificity test was calculated by using PASS software (PASS 11 citation: Hintze J (2011). PASS 11. NCSS, LLC. Kaysville, Utah, USA).

http://applications.emro.who.int/emhj/V17/07/17_7_2011_0582_0586.pdf?ua=1

Sample size calculation for sensitivity and specificity analysis for prevalence of disease from 5% to 20%.

Please Note The Figure Below

4. The authors have not adequately highlighted the strengths and limitations of their study. I suggest better specifying these points.

The fact that curettage operations have limitations in the diagnosis of endometrial polyp and other pathologic conditions indicates the need for a minimally invasive and the most accurate method like hysteroscopy for the evaluation of the uterine cavity in women with PMB. Also, TVS is unreliable as there is subendometrialedema, which makes it difficult to get an accurate measurement of the true ET.

5. What are the actual clinical implications of this study and what does it add to the literature already available on the subject? Indeed, it is important to report the results obtained by the authors in the context of clinical practice and to adequately highlight what contribution this study adds to the literature already existing on the topic and to future study perspectives.

Well, it's worth mentioning that clinical implications is highly essential, and it will be more investigated and need to be further evaluated and scrutinized  in separate and further upcoming research

6. I could not find any information regarding the approval of the Institutional Review Board. Did author this approval before the study start?

The researcher gained the approval of the Institutional Review Board (IRB), JUST, JORDAN, before the study start.

7. I could not find any information regarding the informed consent of enrolled patients. Did author obtain informed consent for each patient? Conversely, this point may raise serious concern from the ethical point of view.

The researcher gained the ethics approvals from The Ethics and Scientific research Committees in Mu’tah University (Reference Number: 20137, 19/02/2013).

8. Previous published data suggested a potential difference of endometrial pathologies, especially endometrial cancer, in women with breast cancer treated with tamoxifen (TAM), aromatase inhibitors (AIs), or receiving no treatment (NT). Nevertheless, recent pieces of evidence suggest that TAM treatment does not seem to be associated with a higher rate of endometrial cancer in women with breast cancer compared with women treated with AIs or NT. I invite authors to discuss (at least briefly) this point, referring to: PMID: 31425735; PMID: 23599784.

Would You Please Check The ATTACHED LINK: PMID: 31425735; PMID: 23599784.

AND KINDLY CITE THIS ARTICLE

9. I suggest discussing the recent innovations in the field of hysteroscopy and their clinical applications to endometrial biopsy. You may refer to: PMID: 31884079; PMID: 28108938.

Would You Please Check The ATTACHED LINK: PMID: 31884079; PMID: 28108938.

AND KINDLY CITE THIS ARTICLE

Reviewer 2 Report

Comments to authors:

  1. I miss the means of the statistical evaluation of the study.
  2. Could the authors specify the correlation of these diagnostic findings with the therapeutical approach and effectiveness ?
  3. Do the authors have any results of the screening of the hormonal profile or the CT scan of the patients to increrase the knowledge about them ?  

The article could be published after minor revision.

Author Response

I miss the means of the statistical evaluation of the study

Would you please specify which variables need means?

As the outcome variables of our study are dichotomous variable (TV/US, hysteroscopy) Namely Yes Vs. No.

  4authors should specify the correlation of these diagnostic findings with therapeutically approach and effectiveness ?

Could be!

But the main aim of the study was to evaluate the clinical Sonographicevaluation of Postmenopausal Bleeding followed by diagnostic and/or therapeutic hysteroscopy

6. Do the authors have any results of the screening of the hormonal profile or the CT scan of the patients to increrase the knowledge about them ?  

The requested results were unavailable right now as the study design was in retrospective manner from multicenter locations

Golden standard